# Low Activation of Knee Extensors and High Activation of Knee Flexors in Female Fencing Athletes Is Related to the Response Time during the Marche-Fente

**DOI:** 10.3390/ijerph20010017

**Published:** 2022-12-20

**Authors:** Tae-Whan Kim, Jin-seok Lee, Iseul Jo

**Affiliations:** 1Korea Institute of Sport Science, Seoul 01794, Republic of Korea; 2Department of Physical Education, Daegu National University, Seoul 42411, Republic of Korea; 3Department of Physical Education, Graduate School of Yonsei University, Seoul 03722, Republic of Korea

**Keywords:** fencing, gender, quadriceps, hamstring, electromyography

## Abstract

Reaction time is important to determine the performance of fencing. The purpose of this study was to investigate the reaction time and muscle activity and compare the movement among genders during Marche-fente. Fifteen Korean national Fleuret fencing athletes participated and were instructed to perform Marche-fente. Reaction time was measured with Plug & Play equipment and muscle activation was obtained by electromyography. The male athletes (0.94 ± 0.08 s) were faster than female athletes, who a performance of 1.03 ± 0.05 s. As the knee extensors activation was increased, the movement and response time was shorter (rectus femoris (RF); r = −0.526, *p* < 0.05, vastus lateralis oblique (VLO); r = −0.628, *p* < 0.05). In phase 1, men activated more knee extensors in the dominant leg, whereas the activation of knee flexors was increased to maintain a stable posture in women. Additionally, women used other muscles instead of large muscles such as RF and VLO in phase 2. In conclusion, female athletes activated knee flexors rather than knee extensors when moving the center of mass or generating a greater force. Less use of knee extensors is associated with knee injuries; therefore, exercise which activates knee extensors is required for females.

## 1. Introduction

Reaction time acts as an important factor in determining performance in sports; it is defined as the time it takes for a reaction to begin when an unexpected stimulus occurs. Fencing requires instantaneous responses to continuous changes in the opponent and performing hits with fast and accurate lunges [1]. The reaction time shows differences depending on the proficiency [2], and the faster the reaction time, the higher the accuracy [3,4]. Consequently, a fast reaction time is important to enhance fencing performance. Therefore, training is conducted to reduce the time and move faster [5].

The *Marche-fente* is the most fundamental motion in *Fleuret*. *Marche-fente* consists of a *Marche*, i.e., a move to the center by pushing the hind limb when stepping forward, followed by a *fente*, which is a lunge performed by pushing the hind limb and stretching the front limb [6]. It was found that the more skilled the fencer, the shorter the time to complete the *Marche-fente* [7]. They the time for the front leg to touch the ground during the Marche was minimized due to quickly performing the next movement [8]; however, Kim (2019) observed no difference between excellent and non-excellent athletes [9]. It seems that some fencers deceived the opponent by differing the time interval between the *Marche* and *fente* [10]. However, Kim’s findings supported the previous result that the elite group performed the *fente* in a shorter time. As a result, the rapid velocity of the *fente* is important to shorten the overall time of *Marche-fente*.

However, a rapid movement from the *Marche* is necessary to increase the velocity in the *fente* because of the inertia. In the *Marche*, moving with maintaining a stable posture is significant. The low vertical displacement of the center of mass and quick flexion and extension of the lower extremity were shown during *Marche* [11]. Tibialis anterior was activated to sustain the posture, and then knee extensors and ankle plantar flexors of the rear leg were mainly used to step forward [12].

Knee extensors and plantar flexors are also important in *fente*. According to previous research, the high speed and force are determined by the rear leg [13], and rapid extension of the hip and knee joints of the hind leg increases the velocity of displacement [10]. Therefore, the quadriceps of the rear leg are greatly activated in the propulsion, which is significantly connected with the speed of the *Marche-fente* [12]. In addition, the gastrocnemius plays a crucial role in transmitting the force generated in the proximal joint, and in switching the rear leg’s propulsion section and the front leg’s braking section [12]. In a recent study, the association between the gastrocnemius and the horizontal force of the hind leg during a lunge was reported [14]. Therefore, we expected knee extensor and gastrocnemius activation to be key during *Marche-fente*.

However, different patterns of muscle activity were shown between gender. Females performed squats or lunges using the quadriceps rather than the hamstring [15]. Moreover, they activated the lateral quadriceps rather than the medial, which induced the knee valgus, which is a risk factor for knee injuries [16]. Lower extremity injuries occur the most in fencing, with a higher proportion of women [17]. Therefore, we hypothesized that muscle activation between men and women would differ during *Marche-fente*.

We compared the muscle activity of male and female athletes in *Marche-fente*. In addition, we aimed to identify the main factors for performance improvement through correlation and provide them as fundamental data for training.

## 2. Materials and Methods

### 2.1. Participants

In this study, 15 Korean national *Fleuret* athletes, including 8 male athletes (age: 28.8 ± 3.4 years, career: 16.8 ± 3.3 years, height: 177.0 ± 7.3 cm, body weight: 76.3 ± 9.2 kg) and 7 female athletes (age: 29.9 ± 4.6 years, career: 13.9 years, height: 163.3 ± 4.7, body weight: 56.8 ± 5.5 kg), voluntarily participated. Male athletes were taller than female athletes; there were no significant differences in other information. The details of the subjects are shown in Table 1.

### 2.2. Experimental Procedure

The Plug & Play equipment developed by the Korea Institution of Sports Science (S K Kil et al., 2016) was used to measure the reaction time. All subjects were instructed to perform the *Marche-fente* after a random LED signal was given in the reaction time measuring device. A distance sufficient to move was secured and four force plates (Kisler, Winterthur, Switzerland; sampling rate of 1000 Hz) were installed. The maximal ground reaction force (GRF) was divided by the individual’s body weight. An A/D converter was used to synchronize the force plates and the device measuring the reaction time. Subjects were conducted in three trials, and the average value was used in this study.

The analysis phase was set as shown in Figure 1 to measure the movement time during the *Marche-fente*. This was defined as the time from when the LED light was turned on (Event 1), to the time when the foot of dominant leg took off from the ground (toe-off, Event 2), and the time when the fencing knife stabbed the target (Event 3). The section between each event was analyzed by setting Phase 1 (Event 1–2) and Phase 2 (Event 2–3). Reaction times were calculated as the time taken from Event 1 to Event 2; movement times were measured as the time taken from Event 2 to Event 3. The overall reaction time was the sum of the reaction time and movement time.

#### 2.2.1. Electromyography (EMG)

The left and right rectus femoris (RF), vastus lateralis oblique (VLO), biceps femoris (BF), semi tendinosus (ST), tibialis anterior (TA), gastrocnemius lateral (GL), and gastrocnemius medial (GM) were selected to obtain EMG data during the *Marche-fente*. The skin surface hairs were removed; then, wireless EMG (TeleMyo DTS EMG Sensor, Noraxon Inc., Scottsdale, AZ, USA; sampling rate of 1500 Hz) was attached to the location which referred to the guidelines. The EMG signal was filtered with a bandpass (10~350 Hz) and then full-wave-rectified [18]. After rectification, EMG data were smoothed to the root mean square (RMS) with 50 ms. The EMG signal was set to an average value of each phase by setting the maximum activation of the *Marche-fente* to 100%.

#### 2.2.2. Statistics

A normality test was conducted with the Shapiro–Wilk test before verifying the statistical difference of the reaction time and kinematic factors (e.g., ground reaction and muscle activity) between male and female athletes during the *Marche-fente*. An independent sample *t*-test was proceeded as a variable satisfying the normality, and a Mann–Whitney U test was replaced if not satisfied.

Pearson’s correlation analysis was also performed to confirm the correlation between variables. If the variable did not satisfy the normality, Spearman’s correlation analysis was conducted.

## 3. Results

### 3.1. Response Time

There was no difference between the reaction time of the Korean national male (0.33 ± 0.06 s) and female (0.31 ± 0.03 s) fencing athletes (Figure 2a). The movement time in male athletes was 0.61 ± 0.03 s, which was faster than the female athletes’ movement time (0.73 ± 0.04 s). Likewise, in the response time, male athletes (0.94 ± 0.08 s) were faster than female athletes (1.03 ± 0.05 s). Although no significant correlation was observed between the reaction time and the response time, the movement time, and the response time, there was a positive correlation (r = 0.818, *p* < 0.01) (Figure 2b).

### 3.2. Ground Reaction Force (GRF)

There was no difference between the male and female athletes in the maximal anterior–posterior and medial–lateral ground reaction forces on the non-dominant leg in phase 2 (Table 2). Male athletes (1.71 ± 0.12% BW) were represented more highly than female athletes (1.55 ± 0.11% BW) in the vertical GRF. There was no significant correlation between the vertical GRF (%BW) and male and female athletes’ reaction time and response time.

### 3.3. Muscle Activity

In both male and female fencing athletes, the TA of the dominant leg was greater activated in phase 1 (Figure 3). Male athletes showed the highest activation in the order of TA–RF–ST, whereas female athletes showed the highest activation in the order of TA–ST–BF. In phase 1, men (12.3 ± 0.05%) activated more RF muscle than women (5.6 ± 0.03%). In the non-dominant leg, both groups showed a greater activation of TA, similarly to the dominant leg. In addition, quadriceps (RF and VLO) were highly activated. However, the ST activation of non-dominant legs was lower in male athletes (4.1 ± 0.02%) than in female athletes (12.6 ± 0.08%).

In phase 2, male athletes exhibited high activation of the dominant leg’s ST, TA, and RF, whereas ST, TA, and BF were mainly activated in women. A higher activation of RF in male athletes (35.5 ± 0.05%) was presented than in female athletes (20.5 ± 0.13%). In the non-dominant leg, both groups showed large activations in TA, BF, and ST. However, female athletes (72.6 ± 0.15%) were greater than male athletes (54.1 ± 0.08%) in TA activation.

The activation of RF in the dominant leg was correlated with the response time in phase 1 (*r* = −0.721, *p* < 0.01), and the movement time (*r* = −0.786, *p* < 0.01) and response time in phase 2 (*r* = −0.704, *p* < 0.01). Negative correlation between BF activation and the reaction time was shown in phase 1 (*r* = −0.663, *p* < 0.01).

In the non-dominant leg, the response time was a negative correlation with VLO (*r* = −0.628, *p* < 0.05) in phase 2. On the other hand, a positive correlation was represented between the TA and movement time (*r* = 0.542, *p* < 0.05).

## 4. Discussion

We investigated the difference in reaction time and muscle activation during *Marche-fente* between male and female elite athletes and the correlation between the reaction time and the kinetics of the lower extremity muscles.

According to previous studies, it has been reported that the shorter the height, the faster the reaction time due to neurological transmission and limb inertia [19,20]. Although men were approximately 8% higher than women, the reaction time did not differ between male and female athletes (Figure 2a). In phase 1, the reaction time was 0.33 ± 0.06 s for male athletes and 0.31 ± 0.03 s for female athletes. This was similar to previously reported values in elite fencers, and faster than novices [21].

National fencing athletes reacted quickly when an unanticipated situation was given. However, a faster reaction time was not only important in the elite, but also constant muscle activation patterns and appropriate use were significant in the elite [22,23]. Different activation among genders to lift off was observed in this study.

The front leg moved forward quickly with less chance of vertical displacement in the center of mass was important during *marche* [11]. Kim and Oh (2009) argued that evenly distributing body mass to both legs enabled pushing vigorously with a stable posture and increase the step during the *fente* [24]. Therefore, the TA of the dominant leg was first activated to sustain the posture and take the dominant leg off [12]. This was consistent with the present result that both male and female athletes showed high activation in the TA of the dominant leg (Figure 3). As the dominant leg, the greater activation in TA of the non-dominant leg was also observed (Figure 3). Therefore, TA muscle contributed to balance and stable posture.

However, the most highly activated muscles in phase 1 were different (Figure 3). Except for the TA muscle, men showed greater activation in RF and ST of the dominant leg, whereas women showed greater activation in ST and BF of the dominant leg. TA and RF in the front leg were activated to take off from the ground and move forward before being activated in the rear leg [12]. This was shown in male athletes, but female athletes used less RF muscle when lifting their front leg, considering the low activation in RF (5.6 ± 0.03%).

In addition, RF and VLO of the non-dominant leg greatly contributed to phase 1. Guilhem (2014) observed that knee extensors and plantar flexors of the rear leg were mainly used in the *marche* for displacement, i.e., fencing athletes moved forward with quadriceps. However, the ST muscle in the rear leg was more activated in female athletes than in male athletes in phase 1 (Table 3). Hamstrings are dedicated to the extension hip joint and flexion knee joint at the late phase of the *marche*. Females stepping forward with pre-activated ST in phase 1 should result in a shorter movement time than males. However, the present data contradicted this hypothesis (Figure 2a). ST muscle might be eccentrically activated for stabilization of their start position which generated abduction.

During the *Marche-fente*, male athletes showed a faster movement time (0.61 ± 0.03 s) and response time (0.94 ± 0.08 s) than female athletes (movement time; 0.73 ± 0.04 s, response time 1.03 ± 0.05 s), but not in the reaction time. The movement time and response time of the fencing athletes presented a high correlation (Figure 2b). Therefore, reducing the movement time was important to rapidly perform *Marche-fente*.

A quick movement from the *marche* is necessary to increase the velocity in the *fente* because of the inertia. The rear leg must be fast from the *marche* and pushed long in the *fente* to quickly perform the *Marche-fente* [13]. The quadriceps of the rear leg were highly activated to move forward and increase the speed and momentum. Therefore, the greater activation of knee extensors (approximately threefold) was observed in phase 2. In this study, the response time decreased as the activation of quadriceps (RF and VLO), increased (Table 4). This supported that the quadriceps activation of the rear leg determined the performance.

In phase 2, the front quadriceps were important to move forward with wide steps. However, lower RF activation of the dominant leg and more TA activation of the non-dominant leg was observed in female athletes. The dorsi flexor and front knee extensors are activated to decelerate and stabilize the body in the braking phase. Nevertheless, the greater the TA activation of the rear leg, the longer the movement time (Table 4). Indeed, greater activation was shown in the rear leg than in the front leg during the *marche*. However, insufficient activation of the front RF might not achieve the necessary distance for the rear leg to move, resulting in increased TA activation of the hind leg. The shorter the stride length, the slower the horizontal speed of the *Marche-fente* [11], suggesting that the shorter stride was due to the slower movement time of the female athletes.

Moreover, females could not achieve greater momentum in propulsion (Table 2). Overall, lower extremity muscles were activated during *Marche-fente*, except for the hamstring, which was slightly activated at around 30%. However, almost twofold greater was shown in female athletes. High activation of the hamstring, to as much as RF and VLO, would disturb the generation of momentum. Previous studies have reported that anteroposterior GRF and gastrocnemius are correlated during a lunge, but there was no difference observed in this study. Both male and female athletes showed increased activation in phase 2 compared with phase 1, because phase 2 included the propulsion of the rear leg and the braking of the front leg.

Regardless of gender, the front ST muscle and rear BF and ST muscles were highly activated in phase 2. BF and ST of the rear leg were mainly used to move forward and control the body in the braking phase. During the lunge, men used the BF more for decelerating their bodies, whereas women used the ST [25]. However, increased ST muscle force created adduction, which was a risk factor for knee injuries. Therefore, this supported that women were more exposed to knee injuries than male.

In summary, female athletes showed lower activation of the quadriceps in movements requiring a quick response. This result might be affected in different muscle strength among gender [26]. However, the strength did not fully explain the muscle activation during movement [27]; therefore, training to increase the activation of knee extensors is required. In addition, further research is needed to generalize our result because of the small number of athletes. Nevertheless, our results obtained from national athletes are meaningful to provide a suggestion of future training and information on gender differences in athletes.

## 5. Conclusions

This study attempted to determine the reaction time and muscle activity of male and female national fencing players during the *Marche-fente* movement. Male athletes performed the *Marche-fente* faster than the female athletes, and as the activation of the knee extensors increased, the movement and response time seemed to be shorter. This means that the role of the knee extensors in *Marche-fente* is important.

In phase 1, male athletes moved with high activation of RF in the dominant leg, whereas female athletes showed increased ST activation to maintain a stable posture. Likewise, in phase 2, which required greater force, other muscles were activated instead of large muscles such as the RF and VLO in female athletes.

As a result, female athletes could not use knee extensors when moving the center of mass or generating a greater force than male athletes. Less use of the knee extensors is linked to knee injuries; therefore, exercises to increase their activation are needed.

## Figures and Tables

**Figure 1 ijerph-20-00017-f001:**
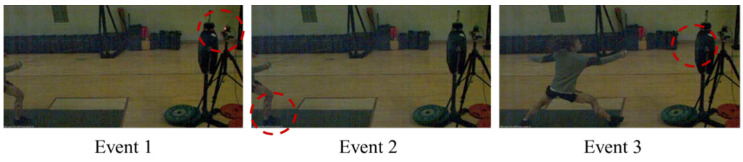
Events during *Marche-fente.* Red circle means that the Led light was turned on in Event 1, and in Event 2 the foot of dominant leg took off.

**Figure 2 ijerph-20-00017-f002:**
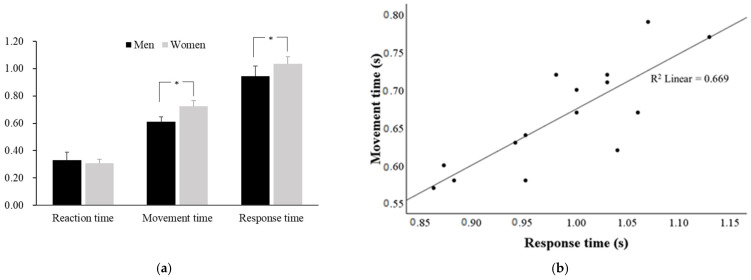
The reaction time, movement time, and response time between male and female athletes (**a**); correlation with the movement time and the response time in fencing athletes including male and female (**b**). * *p* < 0.05.

**Figure 3 ijerph-20-00017-f003:**
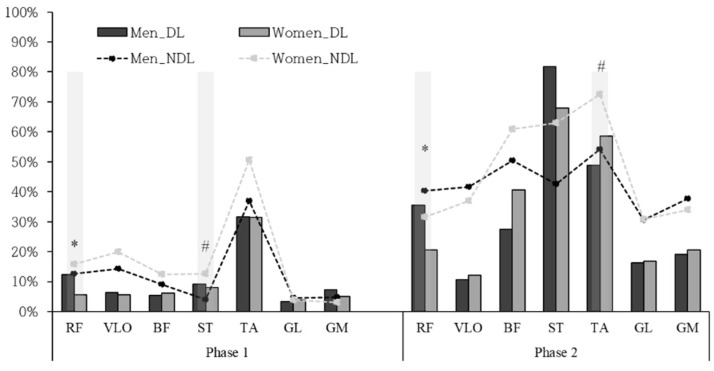
Average muscle activation of dominant and non-dominant leg in fencing national athletes. DL, dominant leg; NDL, non-dominant leg; *, the difference between the male and female athletes in the dominant leg (*p* < 0.05); #, the difference between the male and female athletes in the non-dominant leg (*p* < 0.05).

**Table 1 ijerph-20-00017-t001:** Information of the participants.

Parameter	Male Athletes								Female Athletes						
**Subject**	M01	M02	M03	M04	M05	M06	M07	M08	F01	F02	F03	F04	F05	F06	F07
**Age**	31	29	31	34	29	26	27	23	25	27	30	24	35	34	34
**Career**	20	18	15	20	15	18	18	10	13	7	15	10	15	15	22
**Height**	174.6	189.6	168.0	184.4	175.1	169.0	179.3	176.2	166.7	168.0	156.5	160.1	168.8	160.0	163.0
**Weight (kg)**	72.1	86.2	65.4	84.6	80.6	59.1	80.6	66.7	55.2	58.9	48.7	63.6	64.7	53.0	55.0
**Dominant hand**	L	L	R	R	R	R	R	R	R	R	R	R	L	L	R

R = right, L = left.

**Table 2 ijerph-20-00017-t002:** GRF of male and female athletes (unit: %BW).

Parameters	Male	Female	t	*p*
Peak anteroposterior GRF	0.10 ± 0.04	0.09 ± 0.04	0.46	0.65
Peak mediolateral GRF	0.88 ± 0.11	0.78 ± 0.09	1.84	0.09
Peak vertical GRF	1.71 ± 0.12	1.55 ± 0.11	13.00	0.02 *

* *p* < 0.05.

**Table 3 ijerph-20-00017-t003:** Average muscle activation of dominant and non-dominant leg male and female athletes’ reaction time and response time in phases (unit: %RVC).

Group	Phase	Part	RF	VLO	BF	ST	TA	GL	GM
Male	Phase 1	DL	12.3 ± 0.05 *	6.3 ± 0.02	5.5 ± 0.03	9.2 ± 0.09	31.6 ± 0.07	3.4 ± 0.02	7.3 ± 0.03
		NDL	12.6 ± 0.08	14.4 ± 0.04	9.1 ± 0.05	4.1 ± 0.02 *	37.0 ± 0.08	4.5 ± 0.03	4.8 ± 0.04
	Phase 2	DL	35.5 ± 0.05 *	10.6 ±0.02	27.5 ± 0.03	81.7 ± 0.09	48.8 ± 0.07	16.2 ± 0.02	19.0 ± 0.03
		NDL	40.3 ± 0.08	41.7 ± 0.04	50.5 ± 0.05	42.5 ± 0.02	54.1 ± 0.08 *	30.6 ± 0.03	37.8 ± 0.04
Female	Phase 1	DL	5.6 ± 0.03 *	5.7 ± 0.02	6.2 ± 0.04	8.1 ± 0.04	31.5 ± 0.15	3.9 ± 0.02	5.1 ± 0.02
		NDL	15.8 ± 0.06	19.9 ± 0.09	12.5 ± 0.06	12.6 ± 0.08 *	50.6 ± 0.21	3.8 ± 0.01	3.1 ± 0.01
	Phase 2	DL	20.5 ± 0.13 *	12.1 ± 0.07	40.7 ± 0.21	67.9 ± 0.28	58.5 ± 0.19	16.8 ± 0.05	20.6 ± 0.04
		NDL	31.6 ± 0.08	36.9 ± 0.12	61.0 ± 0.33	63.0 ± 0.39	72.6 ± 0.15 *	30.7 ± 0.09	34.0 ± 0.10

Dominant leg (DL): Front leg when preparing the *fente* with a foot in the same direction as the dominant hand. Non-dominant leg (NDL): Rear leg when preparing the *fente* with foot opposite to the dominant hand. * *p* < 0.05.

**Table 4 ijerph-20-00017-t004:** Correlation between muscle activation of dominant leg and non-dominant leg and reaction time, movement time, response time, and vertical GRF.

Phase	Parameters	DL					NDL				
		RF	VLO	BF	ST	TA	RF	VLO	BF	ST	TA
Phase 1	Reaction time	−0.131	−0.223	−0.663 **	−0.166	−0.093	−0.178	−0.383	−0.217	−0.250	−0.278
	Response time	−0.721 **	−0.319	−0.247	−0.013	0.128	0.199	0.100	0.181	0.289	0.353
Phase 2	Movement time	−0.786 **	−0.218	0.228	−0.083	0.473	−0.511	−0.411	0.086	0.310	0.542 *
	Response time	−0.704 **	−0.392	0.019	−0.003	0.414	−0.452	−0.628 *	0.224	0.131	0.505
	Vertical GRF	0.475	0.343	−0.307	0.282	−0.219	0.345	0.060	0.043	−0.193	−0.171

* *p* < 0.05, ** *p* < 0.01.

## Data Availability

The data presented in this study are available on request from the corresponding author. The data are not publicly available due to ethical restrictions.

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
