# Peer review of "Low Activation of Knee Extensors and High Activation of Knee Flexors in Female Fencing Athletes Is Related to the Response Time during the Marche-Fente"

_ijerph, 2022, doi:10.3390/ijerph20010017_

Round 1

Reviewer 1 Report

Interesting subject of the article. 

I understand the limitations in the size of the groups due to the high sports level of the respondents.

Table 1. It would also be worth mentioning the height of the participants. Reaction time and movement time depend on the length of the nerve route and limb inertia - the longer the limbs, the greater the moment of inertia and the longer the nerve pathways. With large variations, it may affect the response time results.

88 - please specify whether Event 2 is the time when the take off movement begins, or the moment when the foot loses contact with the ground

103-107 - Please describe the statistics methods in more detail.

103-104 please re-edit the sentence as it can be understood that the t-test is used to compare different variables with one another, and not between groups.

Pearson's correlation can be applied, although ANOVA would be better

Both of these tests can be used for the normal distribution - there is no mention in the text as to whether the data meets the normality requirements.

I have no comments to the main text.

I'm just wondering if we analyze reaction time and movement time taking into account the work of muscles, shouldn't we also investigate muscle strength in the conditions of statics and isokinetics at high speed? These data could explain the observed differences between men and women.

Author Response

Dear Reviewer:

First of all, we appreciated the referees' comments with interest in our research “Low activation of knee extensor and high activation of knee flexor in Female Fencing Athletes related to response time during Marche-fente”. We hope to accept our manuscript in an International Journal of Environmental Research and Public Health.

Sincerely,

Iseul Jo

Reviewer 2 Report

The title of atticle:

Low activation of knee extensor and high activation of knee flexor in Female Fencing Athletes related to response time during Marche-fente

The Journal: International Journal of Environmental Research and Public Health

Dear Authors,

At the beginning, I would like to congratulate the action on the basis of very interesting research. This topic is very helpful in the field of fencing, which makes a good contribution to the field of high-skilled sport.

However, I have a few questions before I undertake the effective application of the work to develop publication procedures:

1. Introduction:

- the term "the Marche-fente" very well explained - lines 35-37

- a rich review of professional literature (17 items) has been used throughout the Introduction

- lack of specification of the purpose and research hypotheses in the final phase of this section, or in the Methods section

2. Materials and Methods - lines 70-75

2.1. Participants

- lack of relevant information on the age and experience of the surveyed players in fencing - foil (due to the fact that it is divided into gender: women and men). Missing issues may be crucial for the obtained research results.

2.2. Experimental procedures

- the nomenclature "WEIGHT" is not used in the written text, I suggest changing it to "BODY WEIGHT" in the entire article - line 83

- Were all 3 samples analyzed - by measuring the average values? – line 84

- from what moment was the EMG analysis divided into phase 1 and phase 2 performed? – line 90 and section 3.3. (lines 129-141) - Figure 3. No clear description of phase 1 and phase 2.

1.2.1. Electropyography - line 93

- on what basis was the RMS range - 50ms determined, since there are a lot of possibilities 10-350Hz? What index determines it, or what is the criterion for choosing EMG signal smoothing?

1.2.2. Statistical analysis - line 102

- was the data normal before statistical tests were performed (Pearson's t test and rank correlation)? What criterion/ standard was used to analyze the data when selecting the tests?

2. Results

3.1. Reaction time - line 109

- I suggest using the nomenclature "Reaction time" for "Response time", because it is presented in a given subsection and the topic of the work itself says it

- how exactly was the MT determined, it is not specified? – Figure 2a – line 117-119

3.3. Muscle activity - line 129-141 - repetition of the question:

- from what moment was the EMG analysis divided into phase 1 and phase 2 performed? – line 90 and section 3.3. (lines 129-141) - Figure 3. No clear description of phase 1 and phase 2.

4. Discussion

- the Discussion is relatively rich. The discussion should in the final phase include the defining the goal and the way for its fulfillment and possible solutions for the future.

 In sum:

The sample is reduced, and I understand this because of the level of the fencers (national team), so it seems reasonable to add at the end of this paper that one limitation is the number of the fencers analyzed and for this reason, it is necessary to increase the sample in order to generalize the results achieved.

Lines 256-258

Institutional Review Board Statement: Ethical review and approval were omitted because it was data collected during the training of national athletes. However, consent was obtained for data collection and research.

- In my opinion, any scientific research on humans requires the approval of the Bioethics Committee, even if the research tools used (EMG and Kislter platforms) do not cause major interference. due to the participation of national athletes in this research/ tests.

I hope these comments will be useful.

Best regards,

Katarzyna Piechota, PhD

Author Response

(The authors gave the same response as above.)

Round 2

Reviewer 2 Report

Dear authors,

Thank you for the dynamic work of the authors.

After reviewing the second version of the work after the reviewer's corrections, I believe that the work can be submitted to the next stage of publication procedures.

I accept this version for publication.

Best regards,

K. Piechota, PhD